# Anomaly detection in virtual machine logs against irrelevant attribute interference

**Hao Zhang**[1], **Yun Zhou**[2], **Huahu Xu**[1]\*, **Jiangang Shi**[3], **Xinhua Lin**[4], **Yiqin Gao**[4]

**1** School of Computer Engineering and Science, Shanghai University, Shanghai, China, **2** Shanghai KingLong IoT Co., Ltd., Shanghai, China, **3** Shanghai Shangda Hairun Information System Co., Ltd., Shanghai, China, **4** Shanghai Jiao Tong University, Shanghai, China

\* huahuxu@shu.edu.cn

**Data Availability Statement:** All relevant data are within the manuscript and its Supporting information files.

**Funding:** Shanghai Special Funds for Urban Digital Transformation "O2O Integrated Immersive

## Abstract

Virtual machine logs are generated in large quantities. Virtual machine logs may contain some abnormal logs that indicate security risks or system failures of the virtual machine platform. Therefore, using unsupervised anomaly detection methods to identify abnormal logs is a meaningful task. However, collecting accurate anomaly logs in the real world is often challenging, and there is inherent noise in the log information. Parsing logs and anomaly alerts can be time-consuming, making it important to improve their effectiveness and accuracy. To address these challenges, this paper proposes a method called LADSVM(Long Short-Term Memory + Autoencoder-Decoder + SVM). Firstly, the log parsing algorithm is used to parse the logs. Then, the feature extraction algorithm, which combines Long Short-Term Memory and Autoencoder-Decoder, is applied to extract features. Autoencoder-Decoder reduces the dimensionality of the data by mapping the high-dimensional input to a low-dimensional latent space. This helps eliminate redundant information and noise, extract key features, and increase robustness. Finally, the Support Vector Machine is utilized to detect different feature vector signals. Experimental results demonstrate that compared to traditional methods, this approach is capable of learning better features without any prior knowledge, while also exhibiting superior noise robustness and performance. The LADSVM approach excels at detecting anomalies in virtual machine logs characterized by strong sequential patterns and noise. However, its performance may vary when applied to disordered log data. This highlights the necessity of carefully selecting detection methods that align with the specific characteristics of different log data types.

## 1 Introduction

In recent years, the widespread adoption of virtualization technology has revolutionized modern computing. Virtual machines (VMs) have emerged as a vital element in cloud infrastructure, offering flexibility, resource optimization, and efficient workload management. With the increasing usage of VMs, it becomes imperative to ensure their reliability and security. One critical aspect of maintaining the health and security of virtual systems is detecting anomalies in the extensive and ever-changing logs generated by these machines. These logs provide

Teaching Platform Project based on 5G+AI Data-driven"(No. 202201026).

**Competing interests:** NO authors have competing interests Enter: The authors have declared that no competing interests exist.

valuable information about system activity, playing a crucial role in troubleshooting, performance monitoring, and ensuring the overall integrity of virtualized environments [1]. However, the substantial volume and complexity of these logs, combined with the significant noise present, pose considerable challenges in detecting abnormal patterns or behaviors. Current methods for anomaly detection often rely on clean, labeled datasets, which are rarely available in real-world scenarios. Additionally, many traditional approaches struggle to adapt to the intricate nature of virtual machine logs, leading to reduced detection accuracy. To address these issues, this paper introduces a novel algorithm designed specifically for the anomaly detection of virtual machine logs. Our approach effectively tackles the challenges posed by noise in the data and the difficulty of annotating vast amounts of VM log information. By leveraging advanced techniques such as Long Short-Term Memory (LSTM) networks and Autoencoder-Decoder architectures, the proposed algorithm enhances feature extraction while maintaining robustness against noise. This study fills a critical gap in the current literature on VM logs anomaly detection and provides a solution suitable for real-world applications. Common anomalies can be classified into the following categories.

Anomalous log event sequence refers to a situation where the sequence of events in a log deviate from the expected pattern. For instance, in a typical scenario, a log statement GET should be followed by CHECK, but in an anomalous situation, it may be followed by an unusual event. Various methods can be considered to address this issue. One approach involves using statistical methods such as time series analysis or pattern recognition algorithms. These methods can detect anomalies by identifying deviations from normal log patterns. They are relatively easy to implement and can yield satisfactory results in certain cases. However, they may struggle to detect complex or subtle anomalies that do not adhere to predefined patterns. Another approach is to employ machine learning techniques such as clustering or classification algorithms. This method involves learning patterns from labeled log data and identifying anomalies based on deviations from these learned patterns. One advantage of this approach is its ability to detect both known and unknown anomalies, as it can adapt to new patterns. However, it requires a substantial amount of labeled training data and may be computationally intensive [2].

Log time interval anomaly indicates a significant increase in the execution time of a specific event, typically suggesting a system performance issue. In a normal scenario, the time interval following a GET event should be short, but in an anomalous situation, it takes a considerably longer. One approach to detect such anomalies is the fixed threshold method, where a predetermined threshold is set to determine if a log time interval is considered anomalous. This method is advantageous due to its simplicity and ease of implementation. However, it may not be effective in detecting subtle anomalies or adapting to changing patterns in log data. Another method is the statistical method, which involves analyzing the statistical properties of log time intervals to identify anomalies. This method offers more flexibility and adaptability compared to the fixed threshold method. By considering the distribution and patterns in log data, it can capture both subtle and severe anomalies. However, it may require additional computational resources and expertise to properly apply statistical modeling techniques [3].

Log event parameter anomaly refers to abnormal values of specific parameters in the log. For example, a log event template can have different parameter values, and some exceptional values may indicate abnormal situations. Statistical methods are widely used in current industry, which is consisted of following steps. Firstly, feature extraction can be performed on the log event parameters to capture crucial information. Secondly, statistical methods such as computing the mean, variance, maximum, and minimum values of the parameters can be used to detect outlier values. Additionally, machine learning techniques such as clustering, classification, or anomaly detection algorithms can be utilized to establish models and identify

parameter anomalies. These methods excel at automating the processing of large volumes of log data and uncovering hidden abnormal behaviors. However, they may have limitations in handling complex log structures or high-dimensional parameter spaces, requiring further optimization and improvement. Furthermore, defining parameter anomalies also poses a challenge and needs to be defined and adjusted based on specific circumstances [4]. Apart from the aforementioned types of anomalies, other log features such as log level, process ID, and component can also be used for anomaly detection.

Virtual machine log event sequences essentially represent the occurrence order of execute statements in the source code. Deviation from the normal program sequence often indicates problems with the virtual machine. Therefore, detecting anomalies in event sequences can help identify abnormalities during runtime. The objective of unsupervised learning is to uncover hidden structures from unlabeled data, without the need for any prior knowledge. Virtual machine log data is substantial and dynamic, potentially encompassing various permutations of both normal and anomalous log sequences. Employing unsupervised learning methods alleviates reliance on labeled data. Given the continual evolution of system environments and operating conditions, novel patterns of log sequences anomalies may emerge. Utilizing unsupervised learning approaches facilitates better adaptation to such variations.

In this work, we are motivated by the need to enhance detection accuracy in log event sequences. Our goal is to develop a robust detector capable of accurately identifying anomalies, even in noisy log data. Inspired by these prior efforts, this paper presents an integrated LSTM-AE-based model for anomalous log event sequence detection. The LSTM-AE is applied to learning features following a certain distribution, which are then processed by an SVM for anomaly detection. Specifically, this study aims to enhance the effectiveness of anomaly detection in virtual machine logs, particularly in the context of irrelevant attribute interference. By addressing these challenges, we seek to improve detection accuracy and provide a robust solution for real-world applications.

The schedule for this work is as follows. In Section 2, we reviewed related work. Section 3 provides background knowledge. Section 4 proposes the method, including problem definition and solution design. The details of experimental design and dataset are presented in Section 5. Section 6 and Section 7 displays and discusses the experimental results. Section 8 provides guidance for future work.

## 2 Related work

Log anomaly detection is one of the important applications in the development of intelligent operations and maintenance. It is also a key link in combining machine learning with operations management [5]. Research has shown that applying machine learning to log analysis can effectively solve the analysis and management challenges caused by the expanding volume of data center logs [6]. The task of log anomaly detection generally includes steps such as log collection, log parsing, feature representation, and anomaly detection [7]. The main goal of log anomaly detection is to achieve automated monitoring of the system's operating status and promptly detect and identify abnormal conditions [8]. In the research of log anomaly detection, some traditional machine learning methods have been widely used [9–11]. In addition, log anomaly detection also involves research on semantic representation of logs, online model updating, algorithm parallelism, and generality [2, 13]. Current research focuses mainly on how to achieve automated log anomaly detection and improve the interpretability and decision-making capability of detection results [14].

Various techniques and algorithms are used in log parsing methods. These include clustering methods, frequent pattern mining methods, evolution methods, and log structure heuristic

methods [15]. Clustering methods assume that similar logs belong to the same group and use appropriate string-matching distances for log clustering. Frequent pattern mining methods assume that message types are a set of tags that frequently appear in logs and use frequent item sets to create log message types. Evolution methods use evolutionary algorithms to find Pareto optimal sets of message templates. Log structure heuristic methods use the structural properties of logs to parse them, and the most advanced method is the use of the Drain algorithm [16], which creates a tree structure based on the assumption that words at the beginning of logs undergo little change. Finally, methods based on the longest common subsequence algorithm use dynamic extraction of log patterns from incoming logs.

There are various methods for feature representation and anomaly detection. In terms of feature representation, features such as event count, event sequence, text semantics, time interval, variable values, and variable distributions can be used [17]. The quality of feature representation is crucial for the detection accuracy of subsequent models. In terms of anomaly detection, traditional machine learning methods and deep learning methods can be used. Traditional machine learning methods include principal component analysis, support vector machines, hidden Markov models, K nearest neighbors algorithms, and various clustering algorithms [18–21]. Deep learning methods include long short-term memory networks, bidirectional long short-term memory networks, variational autoencoders, generative adversarial networks, Transformer networks, and CNN [22–30].

In terms of feature extraction, AllInfoLog [31] mentioned four encoders that can extract embeddings of semantics, parameters, time, and other features. As for anomaly detection methods, AllInfoLog mentioned a bidirectional LSTM model based on attention mechanism, which can combine these embeddings for training and outperforms existing log anomaly detection methods in performance and robustness.

In addition, logAD [32] mentioned an integrated learning method that combines multiple anomaly detection techniques to cope with complex log anomaly patterns. This method uses various anomaly detection components, including LSTM-based multivariate time series anomaly detection techniques, distribution distance measurement, and template sequences, to detect different types of anomalies.

GAE-Log [33] comprehensively models logs using event graphs and knowledge graphs. By integrating the temporal dynamics from event graphs and contextual information from knowledge graphs, GAE-Log aims to provide detailed and dynamic representations of log data by considering event sequences and relevant background information from the knowledge repository.

Deeplog [34] focuses on constructing workflows from multiple executions of a single task. Their method's basic idea is 1) mining temporal dependencies between pairs of log keys; 2) constructing basic workflows based on the identified pairwise invariants in the first step; 3) refining the workflow model using input log key sequences. However, they cannot handle log sequences containing multiple tasks or concurrent threads within a task, whereas our research addresses this issue. Table 1 shows comparison of methods for log anomaly detection.

## 3 Preliminaries

First, we provide an introduction to the LSTM and LSTM-AE models. We then elucidate that LSTM-AE can be viewed as a potent fusion of LSTM and AE. Specifically, LSTM-AE employs LSTM for feature extraction from the logs. Subsequently, we utilize the SVM classifier to classify the extracted features.

LSTM, as a replacement for traditional RNN, is designed for time series modeling and overcomes the problem of "vanishing or exploding gradients" in backpropagation when the

**Table 1. Comparison of methods for log anomaly detection.**

| Methods | Challenge I (anomaly detection) | Challenge II (noise resistance) |
|---|---|---|
| He et al. (2016) [9] | Advantages: comprehensive coverage of both log parsing and mining<br>Limitations: weak method's applicability | Uncertain |
| Vaarandi et al. (2003) [10] | Advantages: clustering algorithm tailored for pattern mining from event logs<br>Limitations: weak scalability on large datasets | Weak |
| Grzech et al. (2006) [11] | Advantages: comprehensive coverage of anomaly detection techniques tailored for distributed systems<br>Limitations: weak scalability and adaptability to different network environments | Uncertain |
| He et al. (2018) [12] | Advantages: innovative approach to automating log parsing<br>Limitations: weak accuracy and adaptability | Uncertain |
| Sahoo et al. (2018) [13] | Advantages: adaptation to changing data distributions and environments<br>Limitations: weak scalability and computational efficiency of online learning algorithms | Uncertain |
| Meng et al. (2020) [14] | Advantages: novel approach to incorporating semantic information into log analysis<br>Limitations: complexity and computational cost of implementing the semantic-aware framework | Uncertain |
| Z. Shaeiri et al. (2020) [15] | Advantages: fast and unsupervised detection<br>Limitations: weak accuracy and generalizability | Uncertain |
| He et al. (2017) [16] | Advantages: enables efficient and effective parsing of log data in real-time<br>Limitations: weak scalability and adaptability | Uncertain |
| Chen et al. (2022) [17] | Advantages: effectively capturing temporal dependencies in system log data<br>Limitations: complexity and computational cost | Uncertain |
| Han et al. (2021) [18] | Advantages: strong robustness of anomaly detection<br>Limitations: scalability and computational complexity | Uncertain |
| Paul et al. (2019) [19] | Advantages: effective in internet browsing behavior<br>Limitations: weak generalizability | Uncertain |
| Ying et al. (2021) [20] | Advantages: improving the efficiency and effectiveness of log anomaly detection<br>Limitations: scalability and adaptability to different types of log data and anomaly patterns | Uncertain |
| Lu et al. (2023) [21] | Advantages: dual branch model to enhance the accuracy and efficiency<br>Limitations: complexity of implementing and the requirement for labeled data for training | Uncertain |
| Yang et al. (2019) [22] | Advantages: comprehensive framework for detecting anomalies in log sequences<br>Limitations: complexity and computational cost | Uncertain |
| Han et al. (2021) [23] | Advantages: leveraging natural language-based methods to detect anomalies<br>Limitations: extensive labeled data for training | Uncertain |
| Ryciak et al. (2022) [24] | Advantages: applying natural language processing methods to detect anomalies in log files<br>Limitations: need for robust preprocessing techniques to handle noisy log data effectively | Weak |
| Zhang et al. (2019) [25] | Advantages: robust anomaly detection techniques tailored for unstable log data<br>Limitations: need for comprehensive evaluation on various types of unstable log data | Uncertain |
| Landauer et al. (2018) [26] | Advantages: adaptability to changing system behaviors<br>Limitations: extensive parameter tuning | Sensitivity to noise |

(*Continued*)

**Table 1.** (Continued)

| Methods | Challenge I (anomaly detection) | Challenge II (noise resistance) |
|---|---|---|
| Huang et al. (2020) [27] | Advantages: learn representations at multiple levels of granularity<br>Limitations: need for large amounts of training data and computational resources | Uncertain |
| Hanh et al. (2022) [28] | Advantages: capture both spatial and temporal dependencies in data for anomaly detection<br>Limitations: need for sufficient labeled data to train the model | Uncertain |
| Pan et al. (2023) [29] | Advantages: effectively detect anomalies in various system logs<br>Limitations: need for accurate log template extraction | Sensitivity to noise |
| Gorokhov et al. (2023) [30] | Advantages: can handle uncertainty and imprecision inherent in log data<br>Limitations: computational complexity | Can resist noise |
| Xiao et al. (2023) [31] | Advantages: comprehensive consideration of diverse log features<br>Limitations: computational complexity associated with processing a large number of log features | Can resist noise |
| Zhao et al. (2021) [32] | Advantages: provides real-world validation of various anomaly detection techniques in online service environments<br>Limitations: specificity of the investigated scenarios | Can resist noise |
| Xie et al. (2023) [33] | Advantages: combine adversarial autoencoders with graph feature fusion to enhance robustness<br>Limitations: high computational complexity and difficulty in parameter tuning | Can resist noise |
| Du et al. (2017) [34] | Advantages: can handle large volumes of log data efficiently<br>Limitations: dependency on labeled data and limited interpretability | Can resist noise |

dependency is too long. By incorporating gate units and memory cells into its structure, LSTM can effectively maintain and transmit the key features of data during long-term computations. It can be seen that the LSTM architecture is based on three gate structures, namely the input gate, forget gate, and output gate. The input gate allows information to be stored in each memory cell without disturbance, while the output gate protects other cells from irrelevant disturbance. As for the forget unit, it allows forgetting of irrelevant information [35]. In this study, LSTM will demonstrate its powerful ability to learn features from the logs.

LSTM Autoencoder is an autoencoder model based on Long Short-Term Memory (LSTM) networks. The LSTM network is used to learn the feature representation of input data, which is then reconstructed into the original data through a decoder network. LSTM Autoencoder is commonly used for the representation and reconstruction tasks of sequence data. It can automatically learn the long-term dependencies in sequences and extract key feature information [36].

The output vector $Y_t$ of the gate unit is used to determine the hidden state $h_t$, of the LSTM. The formula is as follows

$$Y_t = f_o(w_o x_t + w_o h_{t-1} + b_o)$$

The formula for calculating the hidden state $h_t$ is

$$h_t = Y_t f_h(C_t)$$

The auto-encoder consists of an encoder and a decoder as illustrated in Fig 1.

The encoder transforms the input $x_t$ into a hidden representation $y_t$ (feature code) using a deterministic mapping function, usually an affine mapping function combined with non-

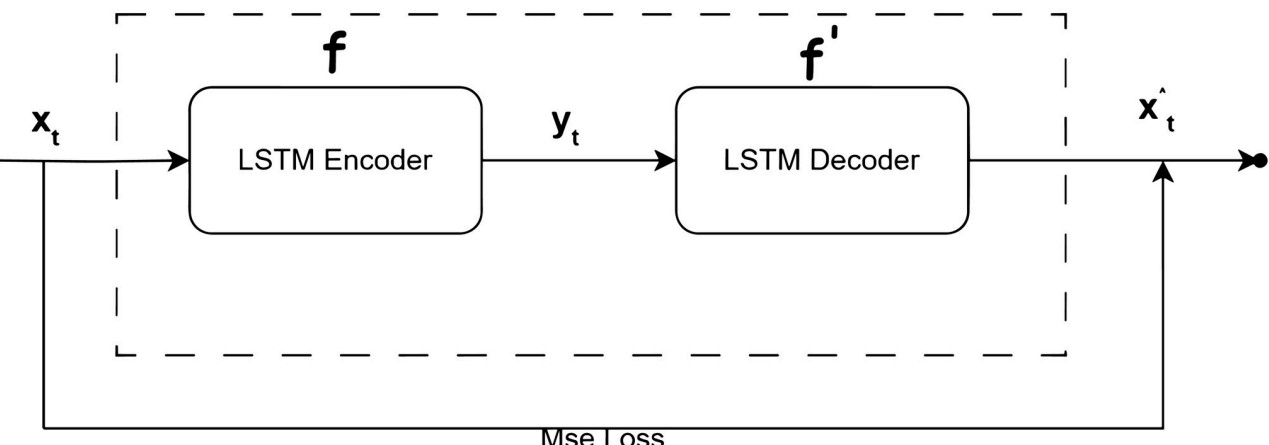

**Fig 1. LSTM auto encoder algorithm illustration.**

linear operations

$$y_t = f(Wx_t + b)$$

Where W is the weight between input $x_t$ and hidden representation $y_t$, and b is the bias.

The decoder implements the reconstruction of the output $\widehat{x}_t$ from the input $y_t$ using the following formula

$$\widehat{x}_t = f'(W'y_t + b')$$

In this equation, W represents the weights between the hidden representation $y_t$ and the output $\widehat{x}_t$, and b represents the bias. $\widehat{x}_t$ can be considered as the reconstruction of the input $x_t$. By minimizing the reconstruction error, we can train the autoencoder and achieve this by minimizing the following loss function J

$$J = \frac{1}{p} \sum_{i=1}^{p} 2muL[x_t, \widehat{x}_t]$$

The LSTM-AE model combines LSTM network with AE (Autoencoder). This means that the encoding and decoding processes are performed by LSTM. Through LSTM, the encoder extracts features from the numerical vectors of the input logs, while the decoder implements the transformation from feature maps to output. Additionally, the parameters for encoding and decoding operations can be calculated using unsupervised greedy training. Unsupervised greedy training is a method used in unsupervised learning, particularly in deep learning models. In this approach, the model is trained layer by layer, with each layer attempting to learn patterns or representations from the data without the need for labeled data. Each layer is trained greedily, focusing only on optimizing its own parameters based on the input from the previous layer. This process allows the model to gradually learn more complex features or representations from the raw data in an unsupervised manner.

Support Vector Machines (SVMs) have been widely used in classification research and have the advantage of automatic complexity control to avoid overfitting. SVMs are developed from learning theory and the main concept is to find a hyperplane in high-dimensional space by maximizing the minimum distance between the hyperplane and the training samples of each class or separating them. Initially designed for binary classification, a one-vs-one SVM training

scheme was proposed to handle multiclass classification problems. Multiclass classification problems can be decomposed into several binary classification problems, and a voting strategy is introduced. Each binary classification is treated as one vote, and the class with the most votes determines the category of the sample.

The definition of SVM is as follows [37]

$$f(y_t) = sign[\omega_s^T \varphi(y_t) + b_s]$$

In order to find the maximum geometric margin $\widehat{\gamma}$, the following optimization problem is proposed

$$\max_{\widehat{\gamma}, \omega_s, b_s} \quad \frac{\widehat{\gamma}}{\|\omega_s\|}$$
$$\text{s.t.} \quad Z_t^i(\omega_s^T y_t^i + b_s) \geq \widehat{\gamma}, i = 1, \dots, m$$

So, we can construct a Lagrangian function to solve the following optimization problem

$$\mathcal{L}(\omega_s, b_s, \alpha) = \frac{1}{2}\|\omega_s\|^2 - \sum_{i=1}^m \alpha_i \left[ Z_t^i(\omega_s^T y_t^i + b_s) - 1 \right]$$

The SVM classifier can be expressed as

$$f(x) = sign\left[ \sum_{j=1}^m \alpha_i Z_t^i K(y_t, y_t^i) + b_s \right]$$

In this equation, "sign" represents the sign function, $\alpha_i$ refers to the Lagrange multiplier, $y_t$ represents the class label of the sample, $K(y_t, y_t^i)$ denotes the kernel function, and b represents the bias term.

Support Vector Data Description (SVDD) is a kernel method used for outlier detection and noise reduction in data analysis. SVDD describes the data by constructing a hyper-sphere, where the sample points inside the hyper-sphere are considered normal points, while the sample points outside the hyper-sphere are considered as outliers or novel points. SVDD has excellent performance in outlier detection and noise reduction, and it is also widely used in support vector clustering and classification [38].

## 4 Method

### 4.1 Formalization of the problem

For Fig 2, it represents a typical fragment of a log file where log lines are displayed in chronological order. Each line serves as the smallest object of our study. For Fig 3, it provides an abstract representation of the general style of a log file, which consists of several lines. Each virtual machine generates a log at a certain time, and the inter-arrival time of log generation varies. Fig 4 shows a log file $T_x$, which is assigned to the normal class, and a log file $T_p$, which is assigned to the anomaly class. A normal log file $T_3$ containing noise is also included and assigned to the anomaly class. Another scenario involves a normal log file $T_3$ containing noise, which is assigned to the normal class Fig 5. However, these classification results raise concerns about (i) how to avoid noise interference during classification, and (ii) how to achieve stability and accuracy in practical production operations.

Collection of virtual machine log data is represented as $D = \{L_1, L_2, L_3 \dots L_N\}$, which includes the log file information for all virtual machine instances. Set of virtual machines $V = \{m_1, m_2, m_3 \dots m_z\}$ represents all the virtual machines on platform V. The number of virtual machines is

023-07-05T11:10:52.872Z| vmx| | 1005: Log for VMware ESX pid=48514369 version=7.0.2

build=build-17867351 option=Release （Line 5）

2023-07-05T11:10:52.872Z| vmx| | 1005: The host is 64-bit. （Line 6）

2023-07-05T11:10:52.872Z| vmx| | 1005: Host codepage=UTF-8 encoding=UTF-8 （Line 7）

**Fig 2. A concrete example showing a few log lines from a VMWare log file.**

denoted as z. $D_t = \{L_1^t, L_2^t, L_3^t \ldots L_Z^t\}$ represents the set of the latest log files of all virtual machines at time t. $|D_t| = z$. Each log data row $\overset{\sigma}{\mu_\lambda^t}$ represents the $\sigma$-th log event in the latest log of the $\lambda$-th virtual machine at time t. $L_\lambda^t = \{\overset{1}{\mu_\lambda^t}, \overset{2}{\mu_\lambda^t}, \overset{3}{\mu_\lambda^t} \ldots \overset{\rho}{\mu_\lambda^t}\}$ represents all the log content in $L_\lambda^t$, where $L_\lambda^t$ has $\rho$ lines of logs in total. A log event, or log line, is also referred to as a log entry.

An anomaly state is represented as $S_x^{t+1} = 1$, indicating the anomaly behavior or state of virtual machine x during its operation at time t+1. $S_x^{t+1} = 0$ represents the normal operation of virtual machine x at time t+1. Dataset $\Gamma_t = \{L_{a1}^t, L_{a2}^t, L_{a1}^t \ldots L_{a\omega}^t\}$ is a set of log files, where $L_{ax}^t$ represents the log file of ax at time t. If $L_{ax}^t$ is in this set, it indicates the prediction of state S for virtual machine ax at time t+1. $\Gamma_t \subseteq D_t$. The judging function $\vartheta$, for any $L_{ax}^t \in \Gamma_t$, $\vartheta(L_{ax}^t) = 1$ indicates $S_{ax}^{t+1} = 1$. $\vartheta(L_{ax}^t) = 0$ indicates $S_{ax}^{t+1} = 0$.

Dataset $\Psi_t = \{L_{b1}^t, L_{b2}^t, L_{b3}^t \ldots L_{bv}^t\}$ is a set of log files, where $L_{bx}^t$ represents the log file of bx at time t. $\Psi_t \subseteq D_t$, and $\Psi_t \cup \Gamma_t = D_t$, $\Psi_t \cap \Gamma_t = \varphi$. The judging function $\zeta$, for any $L_{bx}^t \in \Psi_t$, $\varsigma(L_{bx}^t) = 1$ indicates $S_{bx}^{t+1} = 0$. $\varsigma(L_{bx}^t) = 0$ indicates $S_{ax}^{t+1} = 1$.

Noise $\eta = \{\theta_1, \theta_2, \theta_3 \ldots \theta_o\}$, where $|\eta| = 0$ indicates no noise. The noise function $\Omega(\overset{\sigma}{\mu_\lambda^t}) = 1$ indicates $\overset{\sigma}{\mu_\lambda^t} \in \eta$, and the noise function $\Omega(\overset{\sigma}{\mu_\lambda^t}) = 0$ indicates $\overset{\sigma}{\mu_\lambda^t} \notin \eta$.

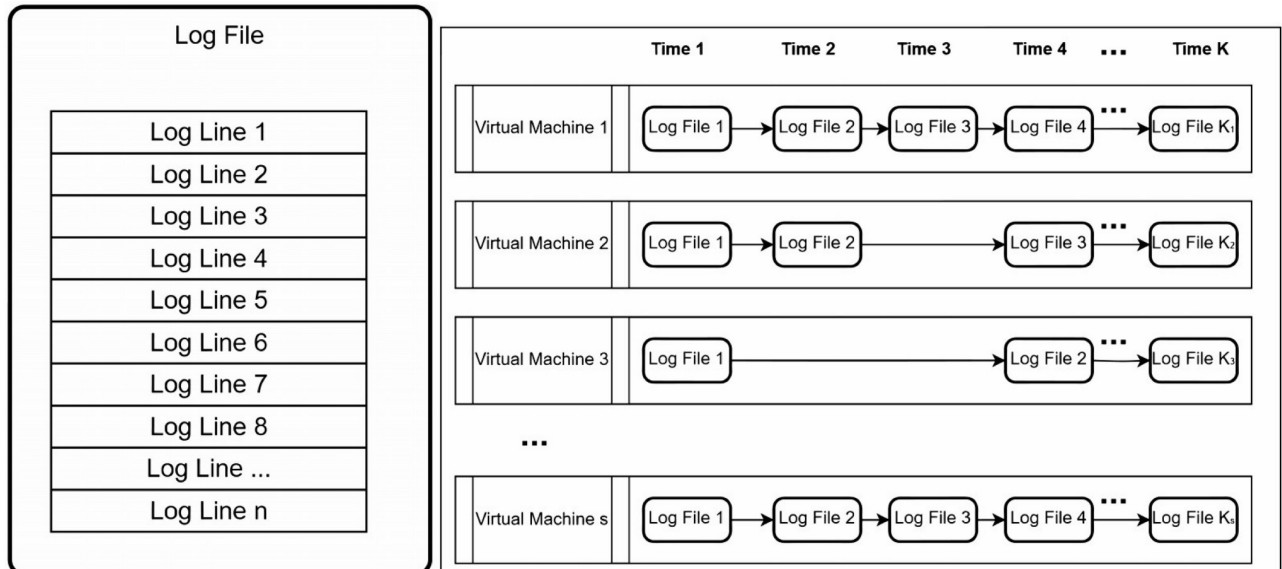

**Fig 3. A schematic diagram of a single log file.** The diagram illustrates the generation of logs. Each virtual machine generates log files in chronological order over time. The intervals between log file generations are often inconsistent, resulting in some virtual machines generating a large number of log files within a given time frame 'K', while others generate fewer log files. The number of log lines in each log file also tends to vary.

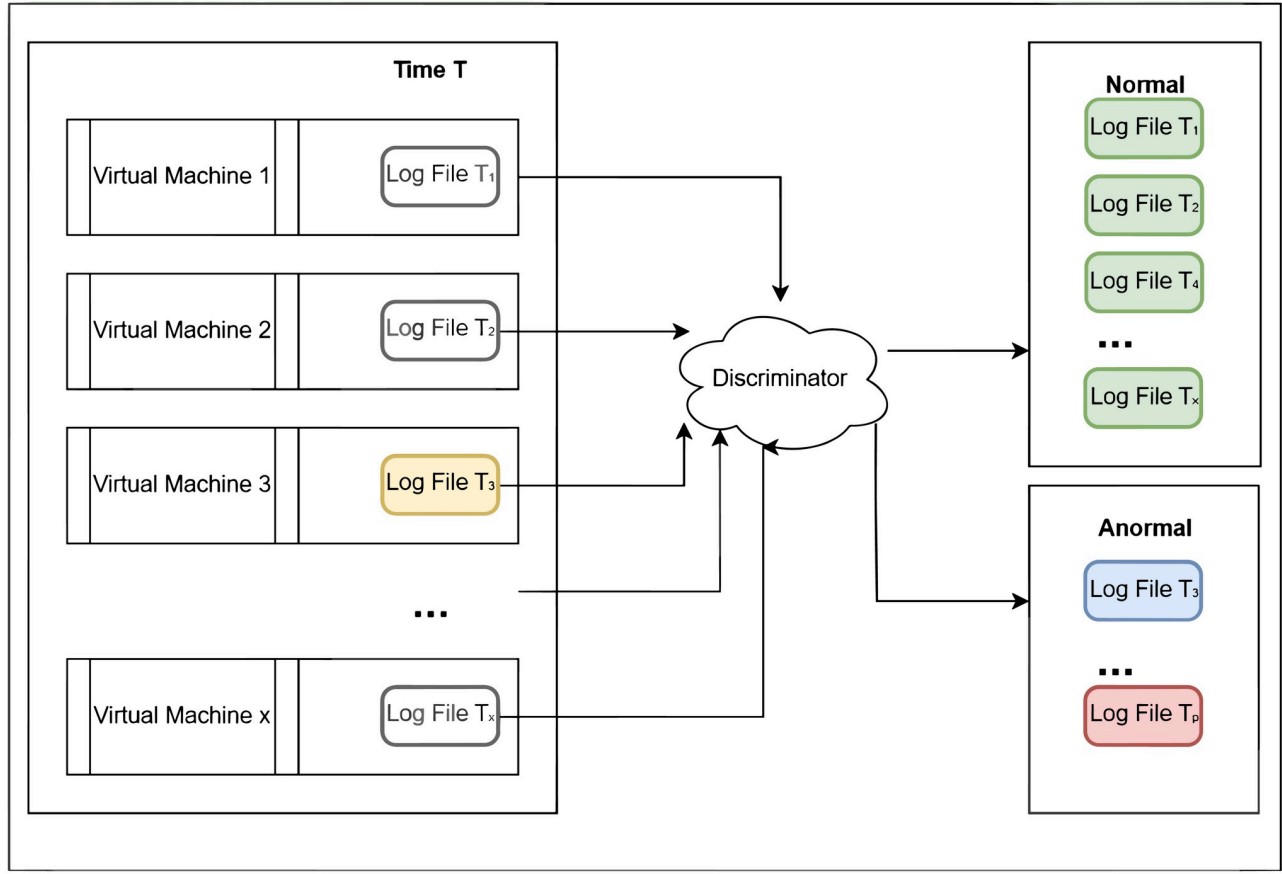

**Fig 4. One case of log file anomaly detection is shown.**

The noise rate $R_t = \frac{1}{\rho}\sum_{i=1}^{\rho}\Omega(\mu_{\lambda}^t)$, represents the noise rate of the λ-th virtual machine at time t, where the noise rate is defined as the proportion of noisy logs among the latest logs events.

For a given $\eta(|\eta| \geq 0)$ and $D_t$, DSC$(D_t) = \{\Gamma_t, \Psi_t\}$. We discuss how to design an early warning system DSC$(D_t) = \{\Gamma_t, \Psi_t\}$ such that $MAX(\sum_{i=0}^{|\Gamma_t|}\vartheta(L_{ai}^t) + \sum_{i=0}^{|\Psi_t|}\varsigma(L_{bi}^t))$.

## 4.2 Overall scheme

The objective of this research is to develop a method for detecting and predicting anomalies based on virtual machine log data. The proposed method follows the following process. Firstly, through analysis of the virtual machine log data, patterns or features that are indicative of abnormal states are identified. Secondly, a set of alert virtual machines is established to associate abnormal states with virtual machine instances that are likely to experience issues. Subsequently, an anomaly detection and prediction system is implemented to monitor the abnormal states of the alert virtual machines in real-time or periodically, with the aim of predicting potential issues in advance and taking preventive measures. Finally, the virtual machine log data set and the alert system are utilized to detect and predict abnormal states of virtual machines, with the ultimate goal of proactively preventing potential problems. The overall scheme is illustrated in Fig 6.

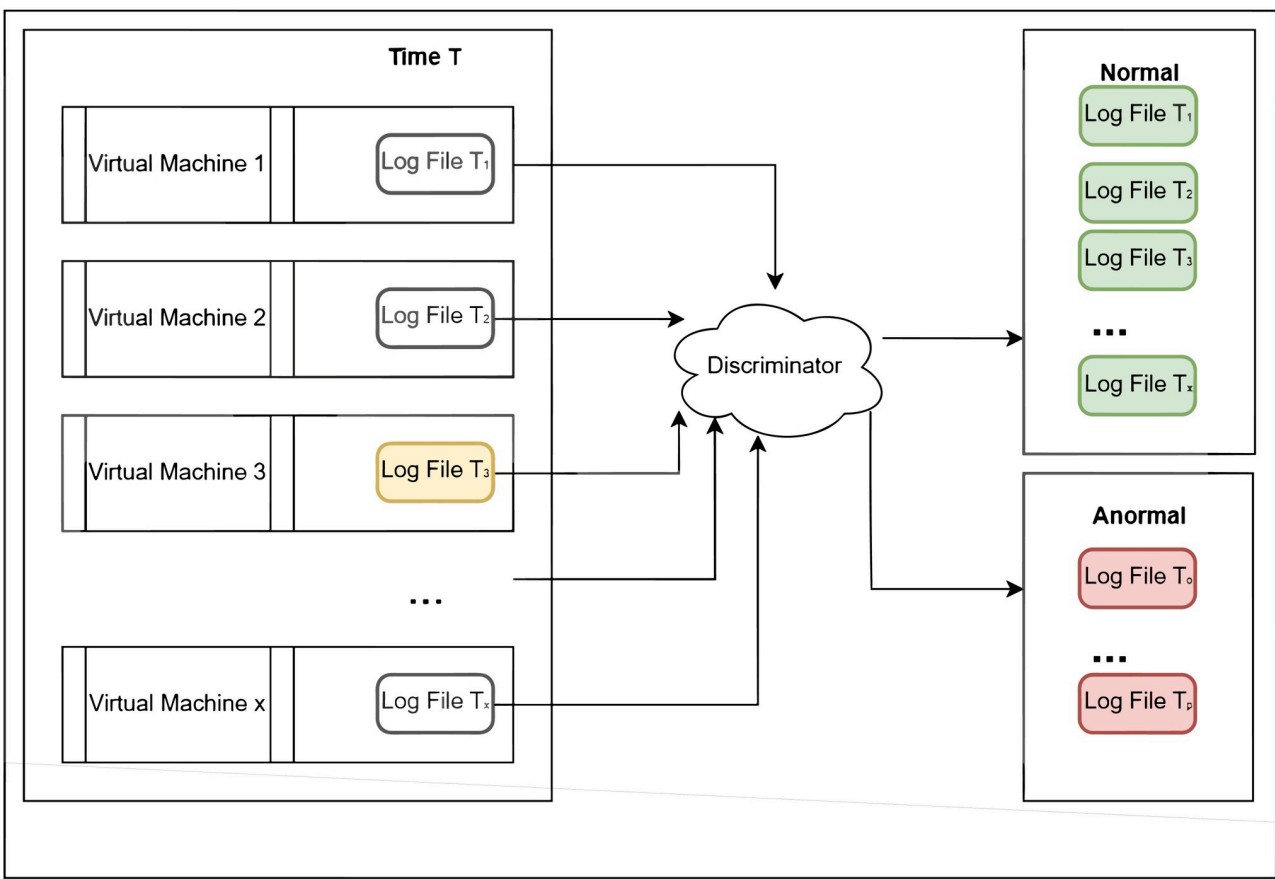

**Fig 5. Another case of log file anomaly detection is shown.** The latest log file on each virtual machine at time T is the object to be detected Discriminator is a detection system. Normal and Anormal represent the two categories into which the log files are divided. In one case (Fig 4), $T_3$ is a noisy normal log file alerted as an anomaly. In another case (Fig 5), $T_3$ is a noisy normal log file considered as normal.

## 4.3 Model implement

The proposed method comprises three primary components: 1) data preprocessing, 2) feature extraction, and 3) anomaly detection. Data preprocessing is implemented in Algorithm 1, while feature extraction and anomaly detection are executed in Algorithm 2. By utilizing a data processing algorithm, we convert the log data into numerical time series data, which serves as the input for the LSTM and AE-based log data feature extraction algorithm. By combining the advantage of LSTM and AE, an LSTM based AE for log anomaly detection algorithm is introduced in this paper, where the LSTM-based AE network is proposed to extract features, and then, an SVM classifier is applied for anomaly detection. More detailed examples for model implementation can be found in the S1 Appendix.

The Drain algorithm is a method used for online log parsing, which effectively and accurately parses raw log messages in a streaming manner. This algorithm does not require source code or any additional information besides the raw log messages. Drain is capable of automatically extracting log templates from the raw log messages and dividing them into distinct log groups. It employs a fixed-depth parsing tree to guide the log group search process, thereby avoiding the creation of excessively deep and unbalanced trees. Moreover, specially designed parsing rules are compactly encoded in the parsing tree nodes. The output of the Drain

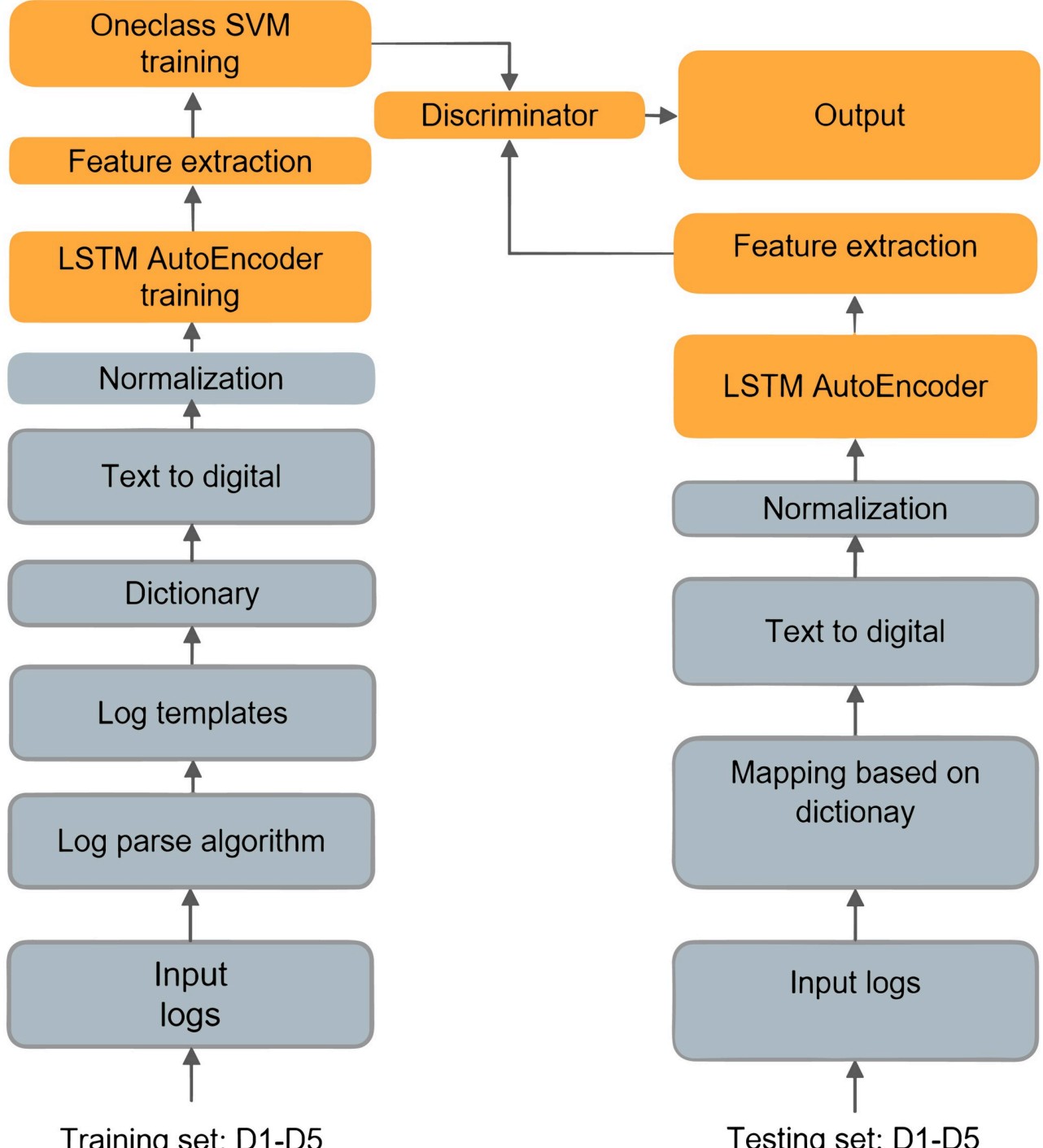

**Fig 6. A brief overview of virtual machine log anomaly detection.** In the training phase, the training log set undergoes log parsing to obtain log templates. The log templates are then sorted based on their length to create a mapping dictionary between the log templates and numerical values. This dictionary converts the log data into numerical data. The feature vector data, obtained through feature extraction, serves as input for training the SVM discriminator. In the testing phase, the log set is mapped into numerical data using the dictionary obtained during the training phase. The feature vector data, obtained through feature extraction, is then used as input for the SVM discriminator to detect anomalies.

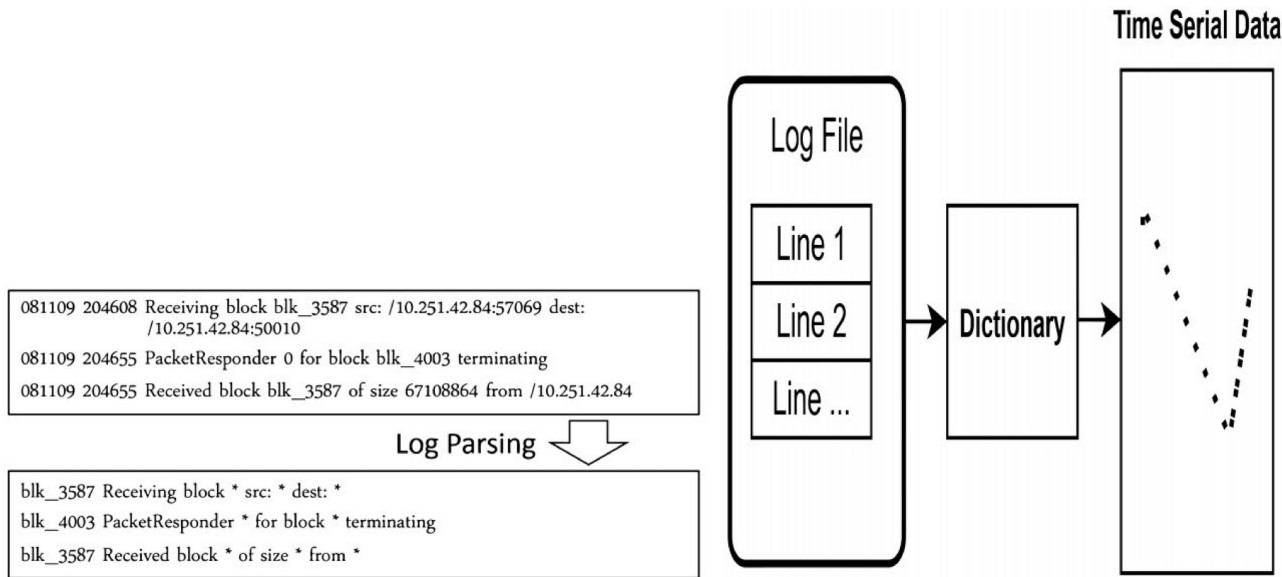

**Fig 7. Data processing diagram, logs are classified and converted into numerical vectors through Algorithm 1.**

algorithm is a data Dictionary. By utilizing this Dictionary transformation, we can convert the log data into time series data as illustrated in Fig 7.

Input is a log sequence $L_k$ containing $\hbar$ log events, each event represented as $\overset{\sigma}{\mu}$, where $\sigma$ represents the index of the event ($\sigma = 1, 2, \ldots, \hbar$).

We get a set of clusters $C$, where each $c_i$ represents a list of event indices, such that each $c_i$ contains similar events satisfying the following conditions

- For all i and j, if $i \neq j$, then $Sim(\overset{i}{\mu}, \overset{j}{\mu}) < T_{sim}$, where $Sim(\overset{i}{\mu}, \overset{j}{\mu})$ is a similarity measure and $T_{sim}$ is the similarity threshold.

- For all i, $c_i$ is non-empty.

Finally output is a time serial vector $V_k$

**Algorithm 1** Data Preprocessing Algorithm

```
Require: Raw log file
Ensure: Preprocessed log vectors
   1: Assign each event μ̇ to its own individual cluster cᵢ.
   2: Initialize C = {c₁, c₂, c₃, ..., cℏ}
   3: while Sim(cᵢ, cⱼ) ≥ Tₛᵢₘ do
   4:    merge cᵢ and cⱼ into a new cluster cₙₑw
   5:    remove cᵢ and cⱼ from C
   6:    add cₙₑw to C
   7: end while

   8: for each log line μ̇ do
   9:    for each cₓ in C do
  10:       if μ̇ match cₓ then
  11:          add x into Vₖ
  12:       end if
  13:    end for
  14: end for
  15: return Vₖ
```

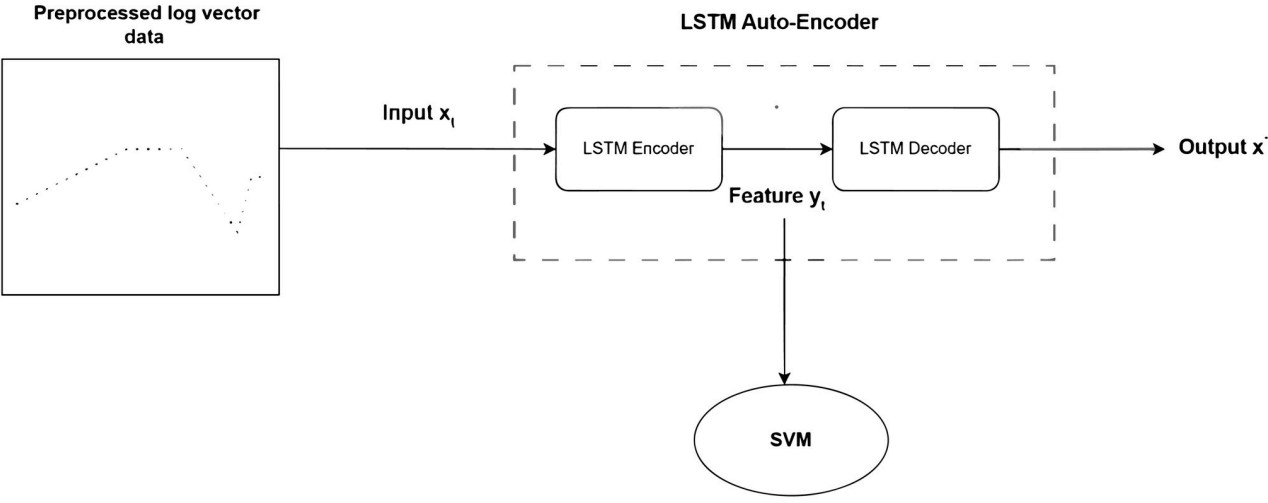

**Fig 8. Overview of Algorithm 2.**

A combined algorithm for feature extraction of log data based on LSTM and AE is proposed. The LSTM-based AE network is used to extract the feature vectors of log data, which are then classified using an SVM classifier. The flowchart of the proposed algorithm is shown in Fig 8. The LSTM-based AE model consists of two LSTM layers, one as an encoder and the other as a decoder. The preprocessed log vector data is used as the input to the LSTM-AE model. The LSTM units are used to extract signal features, and the obtained feature $y_t$ is a probability mapping of length m (m is the output dimension of the LSTM units used as an encoder), representing the feature values ranging from 0 to 1. Then, the LSTM units as the decoder are used to reconstruct the output signal $\widehat{x}_t$. The goal is to minimize the mean square error between the output $\widehat{x}_t$ and the input $x_t$. The smaller the loss function value, the greater the likelihood that the output $\widehat{x}_t$ reconstructs the input $x_t$. This also indicates that the feature code $y_t$ can well represent the input signal $x_t$ in our method. After extracting the features of the log, an SVM classifier is introduced for classification. The input of the SVM classifier comes from the feature $y_t$. SVM classifiers are widely used in pattern classification of feature data and have achieved good results.

**Algorithm 2** LSTM and AE-based Log Data Feature Extraction Algorithm

```
Require: Preprocessed log vector data
Emsure: Classification results
  Initialize LSTM-based AE model
    The first LSTM layer is the encoder.
    The second LSTM layer is the decoder.
2: Train the LSTM-based AE model
    Use the preprocessed log vector data as input.
    The LSTM units in the encoder extract signal features, resulting in
    feature vectors y_t (of length m).
    The LSTM units in the decoder are used to reconstruct the output
    signal x̂_t.
    The goal is to minimize the mean squared error between the recon-
    structed output and the input.
  Feature extraction
The feature vector y_t represents feature values ranging from 0 to 1,
reflecting signal features.
```

```
4: SVM classification
   Use the feature vector y_t as input.
   Apply an SVM classifier to classify different feature vector
   signals.
return Classification results
```

## 5 Experiment

### 5.1 Illustrations of datasetst

In this study, we conducted experiments using virtual machine (VM) log data obtained from 20 virtual machine instances managed by VMware. The dataset comprises 445 independent log files, totaling 1,364,056 log entries, which makes it a large dataset. For the purpose of experimentation, we divided the dataset into five different subsets.

I. Training without noise—Testing without noise (D1). In this setup, we utilized log data without any Gaussian noise for both training and testing.

II. Training without noise—Testing with noise (D2). Here, we trained the model using log data without noise, but intentionally injected 5

III. Training with noise—Testing without noise (D3). In this case, the model was trained on log data that contained 5

IV. Training with noise—Testing with noise (D4). We trained the model using log data with added Gaussian noise and evaluated its performance on similar noisy testing data. There is 5

V. Training without noise—Log sequence disorder—Testing without noise (D5). In this unique scenario, we trained the model using log data without noise, but the log entries were deliberately rearranged.

### 5.2 Assessment metrics and comparison models

#### 5.2.1 Assessment metrics.

$$Precision = \frac{TruePositives}{FalsePositives + TruePositives} \tag{1}$$

Precision evaluates the accuracy of predicting positive log entries. It is calculated by measuring the ratio between true positive predictions and the total positive predictions made by the model. A high precision score indicates that the model minimizes false positive errors.

$$Recall = \frac{TruePositives}{FalseNegatives + TruePositives} \tag{2}$$

Recall, also known as sensitivity, evaluates the ability of a model to correctly identify all positive log entries. It is calculated as the ratio between true positive predictions and the total number of actual positive log entries in the dataset. High recall means that the model effectively captures most of the true positive cases.

$$F1Score = \frac{2 \cdot Precision \cdot Recall}{Precision + Recall} \tag{3}$$

The F1 score represents the harmonic mean of precision and recall. It provides a balanced evaluation of model performance by considering both false positives and false negatives. A

higher F1 score indicates a better balance between precision and recall.

$$\text{MCC} = \frac{TP \times TN - FP \times FN}{\sqrt{(TP + FP)(TP + FN)(TN + FP)(TN + FN)}} \tag{4}$$

Matthew's correlation coefficient (MCC) is a statistical metric used to evaluate binary classification models. With values ranging from -1 to 1, MCC indicates the degree of correlation between actual and predicted classifications. TP stands for True Positives. TN stands for True Negatives. FP stands for False Positives. FN stands for False Negatives.

**5.2.2 Comparison models.**   To comprehensively evaluate our log data analysis method, we have chosen the following comparative models.

1. Neural Network (NN): The neural network is used as a baseline model. These feedforward neural networks consist of multiple layers of interconnected neurons and have been widely used in various machine learning applications [39].

2. Long Short-Term Memory (LSTM) Network: The LSTM network is a special type of recurrent neural network (RNN) known for its ability to capture temporal dependencies in data. It performs well in tasks that require modeling time relationships, making it suitable for log data analysis [40].

3. Support Vector Machine (SVM): SVM is a mature supervised learning algorithm that excels in classification tasks. We choose SVM as a traditional model for comparison because of its robustness and effectiveness [41].

4. K-Means Clustering (KMeans): K-means clustering is an unsupervised learning technique used for data clustering. Although typically used for clustering tasks, we apply KMeans in a unique way for log data analysis [42].

5. Deeplog: DeepLog is a log key-based anomaly detection model and it leverages LSTM to learn the pattern of normal sequence [34].

6. IM: IM mines the invariants among log events from log event count vectors and identifies those log sequences that violate the invariant relationship as anomalies [43].

The selection of evaluation metrics and comparative models aims to comprehensively evaluate the performance of our log data analysis method, taking into account both classification and clustering aspects. This comprehensive evaluation framework ensures a thorough understanding of each model's performance under different experimental conditions.

## 5.3 Experiment description

Experiment 1: The performance of various models will be evaluated and compared under noise-free training and testing conditions using dataset D1.

Experiment 2: The resilience of the models against testing noise will be tested by executing them on dataset D2.

Experiment 3: The resilience of the models against training noise will be tested by executing them on dataset D3.

Experiment 4: The performance of the models in a real production environment with both training and testing noise will be tested by executing them on dataset D4.

Experiment 5: The algorithm's log sequence dependency will be validated by examining if the model effectively captures the relationships between log sequence entries for anomaly detection. This will be done by executing all models on dataset D5.

**Table 2. D1 training without noise—Testing without noise.**

| Model | Accuracy | Precision | Recall | F1-Score | MCC |
|---|---|---|---|---|---|
| LADSVM | 0.9359 | 0.5833 | 1.0000 | 0.7368 | 0.0416 |
| NN | 0.4865 | 0.3897 | 1.0000 | 0.5609 | -0.4734 |
| LSTM | 0.5951 | 0.4812 | 1.0000 | 0.6498 | -0.3388 |
| SVM | 0.3625 | 0.3340 | 0.8571 | 0.4807 | -0.5482 |
| KMEANS | 0.2461 | 0.1690 | 0.6285 | 0.2665 | -0.6197 |
| Deeplog | 0.9358 | 0.6674 | 0.7323 | 0.6983 | 0.3147 |
| IM | 0.8846 | 0.3329 | 1.0000 | 0.4995 | -0.2144 |

Result of experiment 1

The experiments were carried out on a virtual machine with the following specifications: 40 processors, 80GB RAM, 1T hard disk size. The operating system used was CentOS Linux 7.9 v1. All the experiments were conducted in Python programming language using Visual Studio Code as the development environment.

# 6 Results

## 6.1 Noise resistance

Training models on noise-free data is a common practice among engineers in real-world applications. As presented in Tables 2 and 3, our LADSVM model outperforms all baseline models (NN, LSTM, SVM, Deeplog, IM and Kmeans) in F1 score and MCC. This indicates that when using noise-free data for training, our LADSVM model demonstrates strong noise resistance. However, it is worth noting that when the training data is free of noise but the testing data contains noise, the LADSVM model may encounter challenges, leading to weaker performance compared to the results obtained on dataset D1 (as depicted in Table 2). The underlying reason for this discrepancy is that the LADSVM model did not learn how to effectively handle the features associated with noisy data during the training phase. Consequently, this experimental outcome also implies that the performance of the LADSVM model, particularly in terms of F1 score, precision, and recall metrics, can be influenced by the presence of noise in the testing data.

As shown in Table 4, our LADSVM model exhibits the highest accuracy and performs the best in terms of F1 score. Conversely, the comparative NN model excels in recall rate. Additionally, when the training data contains noise but the test data does not, our LADSVM model

**Table 3. D2 training without noise—Testing with noise.**

| Model | Accuracy | Precision | Recall | F1-Score | MCC |
|---|---|---|---|---|---|
| LADSVM | 0.9103 | 0.5000 | 0.5714 | 0.5333 | 0.1861 |
| NN | 0.2807 | 0.2807 | 1.0000 | 0.4383 | -0.7193 |
| LSTM | 0.2451 | 0.2452 | 1.0000 | 0.3937 | -0.7549 |
| SVM | 0.3573 | 0.2869 | 0.9040 | 0.4355 | -0.5812 |
| KMEANS | 0.3589 | 0.2612 | 0.4286 | 0.3245 | -0.3006 |
| Deeplog | 0.5309 | 0.3746 | 0.7323 | 0.4957 | -0.2663 |
| IM | 0.2485 | 0.1662 | 1.0000 | 0.2850 | -0.7440 |

Result of experiment 2

**Table 4. D3 training with noise—Testing without noise.**

| Model | Accuracy | Precision | Recall | F1-Score | MCC |
|---|---|---|---|---|---|
| LADSVM | 0.9231 | 0.5454 | 0.8571 | 0.6666 | 0.0844 |
| NN | 0.3402 | 0.2691 | 1.0000 | 0.4241 | -0.6422 |
| LSTM | 0.4582 | 0.3262 | 1.0000 | 0.4919 | -0.5100 |
| SVM | 0.3273 | 0.2773 | 1.0000 | 0.4342 | -0.6580 |
| KMEANS | 0.3461 | 0.2454 | 0.4286 | 0.3121 | -0.3277 |
| Deeplog | 0.7658 | 0.4805 | 0.7323 | 0.5803 | 0.0016 |
| IM | 0.2410 | 0.1676 | 1.0000 | 0.2870 | -0.7514 |

Result of experiment 3

achieves higher scores in testing compared to the D2 test dataset (where the training data is noise-free but the test data contains noise). This discrepancy can be attributed to the inherent randomness and uncertainty associated with noise, making it more challenging to capture noise features in the test set compared to the training set.

Table 5 demonstrates that LADSVM outperforms in terms of precision and F1 score. This indicates that our LADSVM model can effectively handle noisy datasets and exhibits good adaptability in noisy environments. Comparing the performance of LADSVM on dataset D4 with dataset D2, as shown in Tables 2 and 4, it can be observed that LADSVM achieves higher precision and recall on dataset D4. This can be attributed to LADSVM's ability to identify and discover noise characteristics in the training dataset, which aids in detecting anomalies in the testing dataset within a noisy environment. The inclusion of noise in the training set allows the model to learn specific noise features, thereby enhancing its generalization ability on noisy testing data.

Furthermore, comparing the performance of LADSVM on dataset D4 with dataset D3, as shown in Tables 3 and 4, LADSVM achieves higher precision and lower recall, while the F1-score remains the same. This suggests that LADSVM can adapt to both noisy and noiseless testing environments when trained in a noisy training environment. LADSVM demonstrates better robustness in managing differences in testing conditions.

The differences in performance among various models on different metrics indicate that neural network (NN) and long short-term memory (LSTM) models may be more sensitive to noise in the training data. They have a relatively weaker ability to adapt to noisy datasets and exhibit poorer generalization ability. On the other hand, deep learning methods are adept at accurately capturing the key patterns in noisy data, while non-deep learning methods excel at

**Table 5. D4 Training with noise—Testing with noise.**

| Model | Accuracy | Precision | Recall | F1-Score | MCC |
|---|---|---|---|---|---|
| LADSVM | 0.9359 | 0.6250 | 0.7143 | 0.6666 | 0.2713 |
| NN | 0.3187 | 0.2879 | 1.0000 | 0.4471 | -0.6706 |
| LSTM | 0.4577 | 0.3381 | 1.0000 | 0.5053 | -0.5089 |
| SVM | 0.3387 | 0.2742 | 1.0000 | 0.4304 | -0.6442 |
| KMEANS | 0.3589 | 0.1612 | 0.4285 | 0.2343 | -0.3924 |
| Deeplog | 0.7453 | 0.4995 | 0.7323 | 0.5939 | 0.0082 |
| IM | 0.2414 | 0.1691 | 1.0000 | 0.2893 | -0.7509 |

Result of experiment 4

focusing on robust features in noisy data. This is because deep learning methods typically possess stronger learning and expressive capabilities. By utilizing multi-layer neural network structures, deep learning methods can learn more complex and abstract feature representations, thereby better capturing the key patterns in the data. However, deep learning methods rely on linear and nonlinear transformations of the input data to generate outputs, and in the presence of noise in the input data, the noise is amplified between the layers of the network, thereby affecting the final output. Additionally, deep learning methods may be prone to overfitting, and if the noise data is incorrectly labeled, it may lead to overfitting to the noise. On the contrary, non-deep learning methods are better at focusing on robust features in noisy data due to their typically simpler and more stable nature. Non-deep learning methods may employ fewer parameters and simpler model structures, making them more resilient to perturbations in noisy data. Consequently, they are more likely to disregard small fluctuations in the noisy data and concentrate on more stable and consistent features.

## 6.2 Effects of log sequence

As shown in Table 6, K-means performs the best in terms of accuracy, while LADSVM leads in precision and F1 score. However, our LADSVM model performs significantly weaker on D5 compared to other datasets, indicating its dependence on log order. This is because our LADSVM anomaly detection is based on identifying patterns in log sequences, so when the sequence of log anomalies is disrupted, it degenerates into the SVM method.

The experiment also revealed significant differences in the performance of different models on each metric. In handling structured logs, LSTM (Long Short-Term Memory) and ordinary neural networks usually perform well. These models can learn the sequence and temporal dependencies in the input data and extract important features from structured logs. On the other hand, for handling randomness in the data, K-means and SVM are more suitable. K-means can cluster the data based on the distances between them, thus identifying major patterns or structures when there is randomness or noise in the data. SVM can define a boundary in the dataset to separate the main data patterns from random data and noise.

## 6.3 Comparation of detection perform

As presented in Table 2, the LADSVM model demonstrates superior performance compared to other models across all metrics. This suggests that the model exhibits a well-balanced performance for both positive and negative classes within the dataset. Particularly, the LADSVM model achieves the highest f1 score of 0.7368, indicating its ability to strike a good balance

**Table 6. D5 training without noise—Log sequence disorder—Testing without noise.**

| Model | Accuracy | Precision | Recall | F1-Score | MCC |
|---|---|---|---|---|---|
| LADSVM | 0.2435 | 0.1761 | 1.0000 | 0.2994 | -0.7483 |
| NN | 0.0833 | 0.0816 | 1.0000 | 0.1511 | -0.9166 |
| LSTM | 0.0879 | 0.0812 | 1.0000 | 0.1503 | -0.9116 |
| SVM | 0.1633 | 0.1572 | 1.0000 | 0.2717 | -0.8356 |
| KMEANS | 0.3461 | 0.1638 | 0.4286 | 0.2371 | -0.4012 |
| Deeplog | 0.0863 | 0.0879 | 0.7323 | 0.1570 | -0.8803 |
| IM | 0.113 | 0.0821 | 1.0000 | 0.1517 | -0.8853 |

Result of experiment 5

between precision and recall. Additionally, the accuracy of the LADSVM model is 0.9359, highlighting its proficiency in correctly predicting a significant proportion of samples. Traditional log parsing-based model(IM) performed poorly on datasets D2-D5 because the effectiveness of such method heavily relies on the quality of logs. The presence of noise, errors, or missing information in the logs may lead to decreased accuracy and reliability in invariant extraction.

In the domain of log anomaly detection, the primary objective is to identify abnormal logs. Consequently, metrics like f1 score and accuracy are more appropriate as comparative benchmarks. Figs 9 and 10 illustrate that the NN, LSTM, SVM, Deeplog, IM and KMEANS models exhibit poor performance, as evidenced by their lower f1 scores and accuracy metrics compared to our LADSVM model across various datasets. This suggests that these six models display relatively inadequate predictive capabilities. The experiments conducted have verified the effectiveness of LADSVM in capturing hidden anomalous log sequences within time series log data by combining deep learning techniques with traditional unsupervised learning methods.

## 6.4 Ablation study and training time analysis

We applied LSTM+SVM to each dataset respectively, as illustrated in the Fig 11, the results demonstrate the added value of the autoencoder and decoder components.

The complexity of our algorithm primarily lies in the training phase of the LSTM-based autoencoder model. Initialization incurs minimal cost $O(1)$, while feature extraction is inexpensive $O(1)$. Training complexity, determined by epochs, instances, sequence length, and LSTM units, is approximately $O(k * n * m * d)$. SVM classification complexity depends on support vectors and kernel functions, ranging from $O(n^2 * p)$ to $O(n^3 * p)$ for training and $O(m * p)$ for testing.

We also compared the training time of each model, the results of which are presented in Table 7. The runtime was measured using milliseconds per sequence (ms/seq) as the metric. It can be observed that deep learning models are much slower than non-deep learning models, which is attributed to the higher complexity of deep learning models. The training time of our

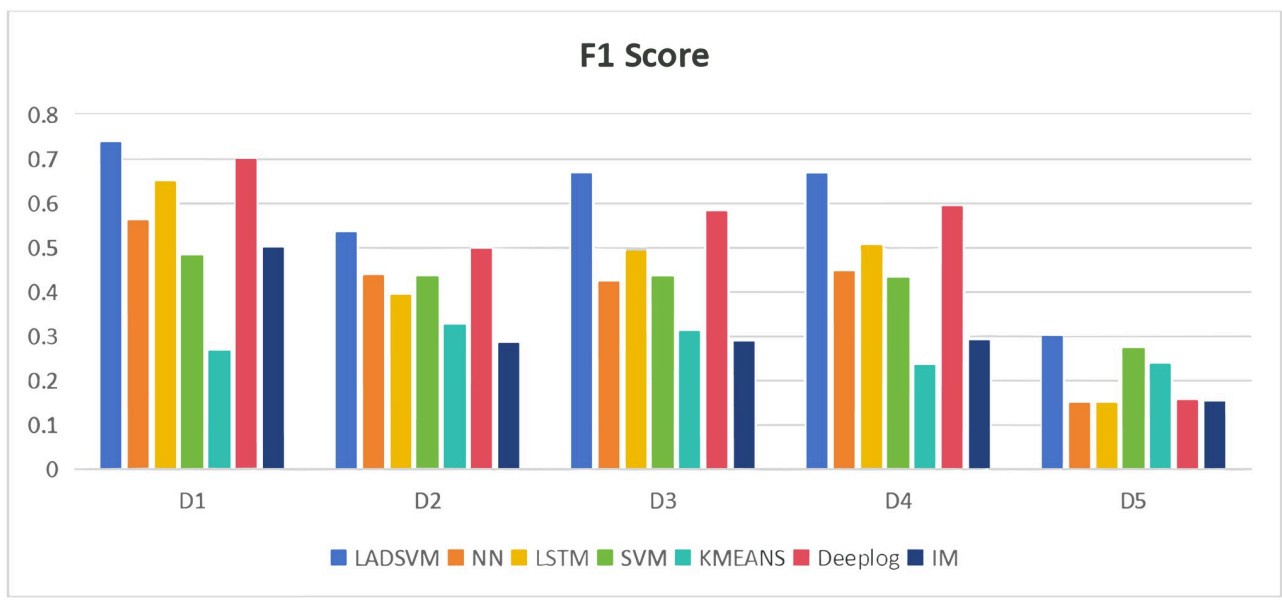

**Fig 9. Comparison of F1 score.**

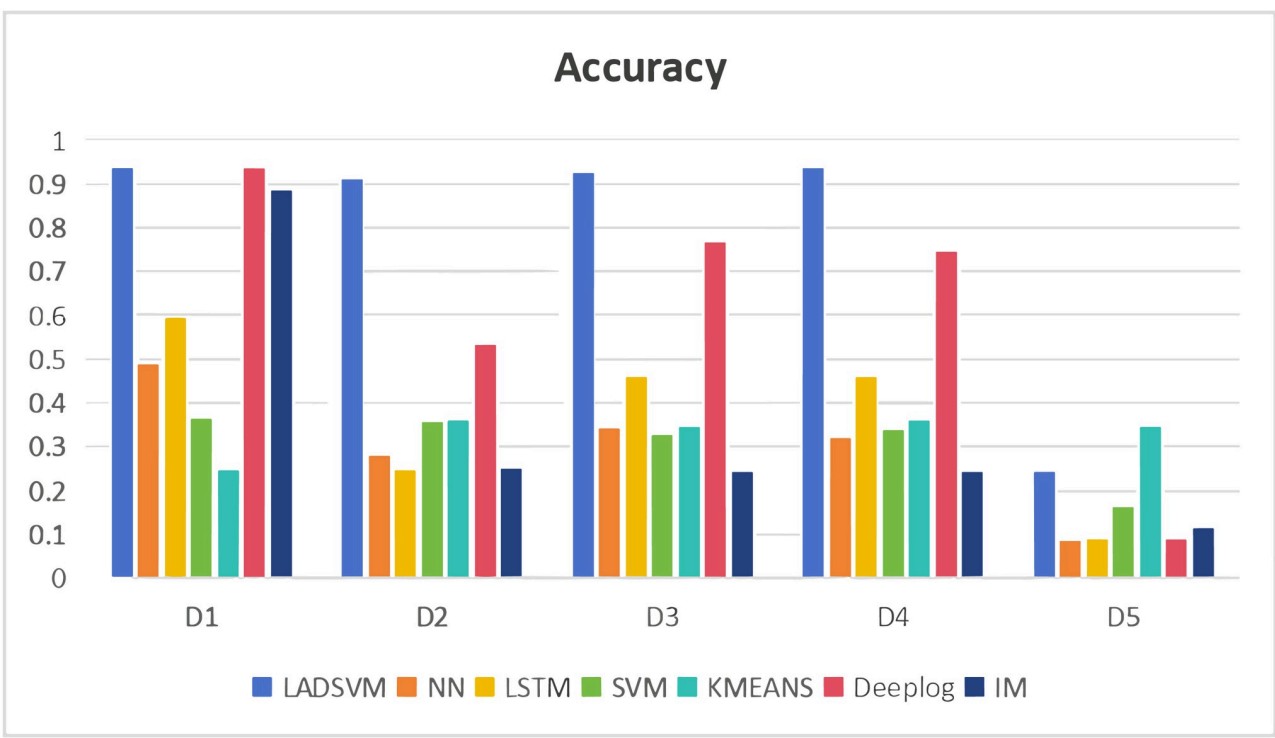

**Fig 10. Comparison of accuracy.**

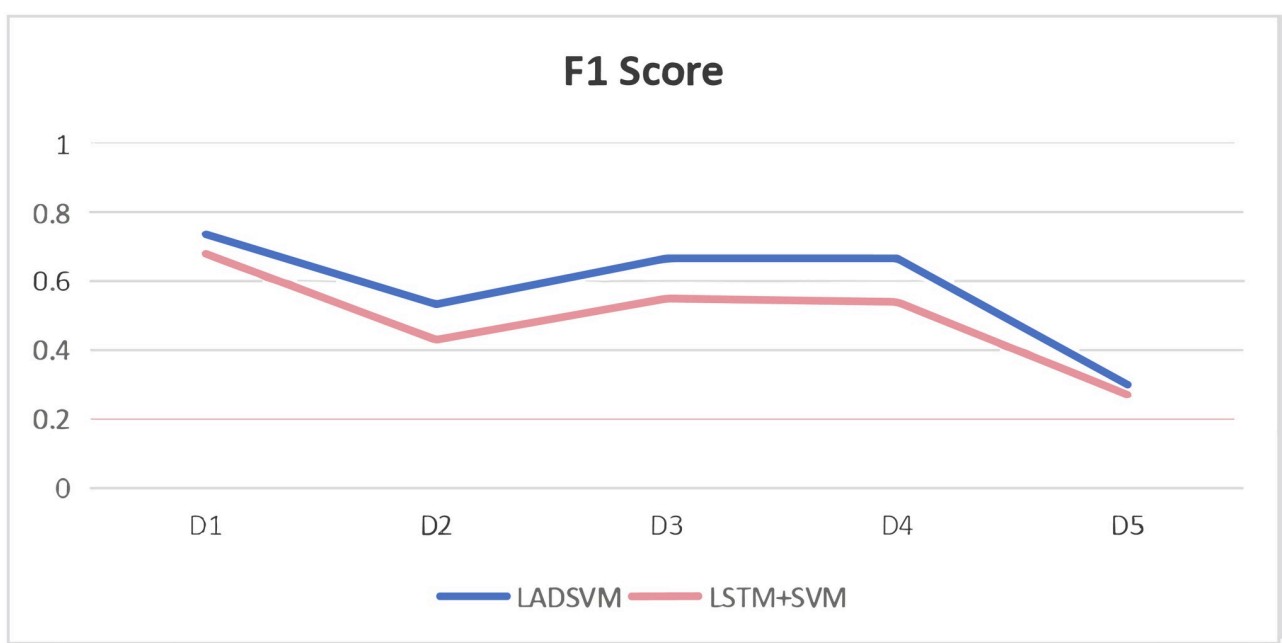

**Fig 11. Ablation study.**

**Table 7. Training time.**

| Model | Training time (ms/log entry) |
|---|---|
| LADSVM | 11.13 |
| NN | 31.97 |
| LSTM | 22.8 |
| SVM | 0.0425 |
| KMEANS | 0.0183 |
| Deeplog | 51.84 |
| IM | 20.95 |

model was shorter compared to other deep learning models. This is because autoencoders typically learn compressed representations of data, reducing the complexity and number of parameters in the network.

# 7 Discussion

## 7.1 Advantage

The LADSVM anomaly detection approach demonstrates significant strengths through its effective handling of large volumes of VM log data, robust noise resilience, and high detection accuracy, making it well-suited for real-world applications. By employing machine learning algorithms and models, LADSVM enables the automatic detection of anomalies in logs, thereby facilitating proactive fault warning and assisting operation and maintenance personnel in event handling. Furthermore, LADSVM is capable of extracting features from log data and learning both normal and abnormal patterns. This empowers the system to identify previously unseen anomalies and adapt to changes in log formats. LSTM autoencoders excel in capturing temporal dependencies and patterns in sequential data, making them effective for feature extraction and representation learning. Their ability to reconstruct input data helps mitigate the impact of noise, as demonstrated in studies such as those by Fatma et al. [44], which highlighted the efficacy of autoencoders in noisy environments. SVMs are powerful classifiers that can effectively separate data points in high-dimensional feature spaces. By combining these two models, we leverage the complementary strengths of each. This combination enables LADSVM to handle a wide range of anomalies. Studies by Zhang et al. [45] have shown that similar hybrid models improve detection accuracy, supporting the robustness of our approach in real-world applications.

## 7.2 Limitation

One limitation of the LADSVM anomaly detection approach is its inability to effectively capture the features of disorder in log data. For instance, in Experiment 5, LADSVM performed significantly worse than on other datasets. Additionally, while the model excels at learning intricate patterns, it lacks interpretability, making it challenging for users to understand how the learned features relate to the original log data. As an unsupervised learning model, LADSVM learns representations without explicit labels, which can obscure the meaning of the detected anomalies. Moreover, the computational costs associated with training deep learning models like LSTM autoencoders can be substantial, potentially limiting their deployment in resource-constrained environments. Finally, as log data volumes increase, ensuring the model's scalability to larger datasets becomes critical for practical applications in large-scale settings.

### 7.3 Insight

There are several important insights and challenges to consider in this context. Firstly, the construction of high-quality feature data is crucial for achieving accurate model detection. The precision of the model in identifying normal and abnormal patterns in logs relies on the quality of the extracted features. Secondly, achieving interpretability of the detection results and providing decision support are essential for log anomaly detection systems. By understanding the features that indicate anomalies, visualizing abnormalities in a way that operators can comprehend, and considering the interdependencies among anomalies, the system can assist in decision-making and enhance the comprehensibility of the detected anomalies. One approach is to adopt large language models at the user end. This approach can assist users in better understanding the anomalous information within the logs and transforming it into user-friendly formats, thereby improving the interpretability of the system and enhancing the user experience. Thirdly, the log sequence disorder primarily stems from errors occurring during the log transmission process or transmission delays. One approach is to establish a comprehensive knowledge base containing various errors and anomalies that may occur during the log transmission process. This would provide the model with rich background knowledge and experiential summaries, aiding the model in better understanding and addressing issues with disorder data. Another approach involves integrating natural language processing techniques to perform semantic analysis of log data, identifying key information and significant features. Additionally, it is important to consider the practical implementation of LADSVM in different real-world environments, such as cloud data centers and industrial IoT systems. In cloud data centers, LADSVM can be utilized to monitor virtual machine logs for anomaly detection and early warning. To enhance the overall anomaly detection capabilities within the data center, LADSVM can be integrated with performance metrics and network log analysis. For example, by monitoring CPU usage, memory consumption, and disk I/O alongside virtual machine logs, the system can detect correlations between performance degradation and log anomalies. Similarly, analyzing network logs for unusual traffic patterns can provide further context for any detected anomalies, allowing for a more comprehensive view of the data center's health. This integrated approach enables engineers to quickly identify and locate anomalies, reducing mean time to resolution (MTTR) and improving operational efficiency. By providing actionable insights and correlating different data sources, LADSVM can significantly enhance the proactive maintenance and management of cloud data center resources. In industrial IoT, LADSVM can also play a pivotal role in monitoring. For instance, in industrial settings, LADSVM could be used to analyze logs from virtualized edge devices that aggregate data from various sensors. By applying LADSVM to these virtual machine logs, it can identify unusual patterns that may indicate operational issues or security threats. Integrating LADSVM with performance metrics—such as equipment status, throughput rates, and environmental conditions—can enhance its ability to detect anomalies that could lead to equipment failures. While LADSVM may not be as directly applicable to traditional industrial IoT device logs, its methodology can still inform the development of tailored anomaly detection solutions for IoT environments. By combining insights from LADSVM with domain-specific approaches, engineers can create systems that monitor both virtualized and physical devices, ultimately improving operational efficiency and reducing downtime. Lastly, there are still unresolved challenges in this field. These include the difficulty of detecting anomalies in log texts with complex semantics and long-distance dependencies using traditional machine learning methods, the need for efficient training of deep models due to their computational demands, and the trade-off between detection accuracy and the size of the time window used for log segmentation.

## 8 Conclusion

In general, this approach addresses the challenge of efficiently identifying abnormal behavior in large volumes of virtual machine logs generated within a virtual machine platform. Collecting abnormal system logs in real-world scenarios makes accurate parsing and anomaly detection a time-consuming task. To overcome these challenges, we introduce LADSVM, which first processes logs using a parsing algorithm, followed by feature extraction through a combination of Long Short-Term Memory (LSTM) and Autoencoder (AE) networks. A Support Vector Machine (SVM) classifier is then employed to categorize the feature vectors. The main findings highlight that our novel deep learning algorithm effectively handles log sequences with multiple tasks or concurrent threads, outperforming traditional methods by learning better features and demonstrating superior noise resistance. Notably, deep learning methods excel at capturing key patterns in noisy data. Looking ahead, future research will focus on log semantic representation, online model updating, algorithm parallelism, and enhancing the interpretability of detection results, which are vital for advancing intelligent operations and maintenance.

## Supporting information

**S1 Data.**
(RAR)

**S1 Appendix.**
(PDF)

## Author Contributions

**Conceptualization:** Hao Zhang, Huahu Xu.

**Data curation:** Hao Zhang.

**Methodology:** Hao Zhang.

**Project administration:** Hao Zhang.

**Software:** Yun Zhou, Jiangang Shi, Xinhua Lin, Yiqin Gao.

**Writing – original draft:** Hao Zhang.

**Writing – review & editing:** Hao Zhang, Huahu Xu.

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
