## [Decision Letter · Decision Letter 0]

22 Oct 2024

PONE-D-24-25514Anomaly detection to virtual machine logs for irrelevant attribute interference: a case study in campus data centerPLOS ONE

Dear Dr. Xu,

Thank you for submitting your manuscript to PLOS ONE. After careful consideration, we feel that it has merit but does not fully meet PLOS ONE’s publication criteria as it currently stands. Therefore, we invite you to submit a revised version of the manuscript that addresses the points raised during the review process.

Please carefully adress all the suggestions and comments raised by the reviewers.

We look forward to receiving your revised manuscript.

Kind regards,

Arne Johannssen

Academic Editor

PLOS ONE

Journal requirements: When submitting your revision, we need you to address these additional requirements. 1. Please ensure that your manuscript meets PLOS ONE's style requirements, including those for file naming. The PLOS ONE style templates can be found at https://journals.plos.org/plosone/s/file?id=wjVg/PLOSOne_formatting_sample_main_body.pdf and https://journals.plos.org/plosone/s/file?id=ba62/PLOSOne_formatting_sample_title_authors_affiliations.pdf 2. PLOS requires an ORCID iD for the corresponding author in Editorial Manager on papers submitted after December 6th, 2016. Please ensure that you have an ORCID iD and that it is validated in Editorial Manager. To do this, go to ‘Update my Information’ (in the upper left-hand corner of the main menu), and click on the Fetch/Validate link next to the ORCID field. This will take you to the ORCID site and allow you to create a new iD or authenticate a pre-existing iD in Editorial Manager. 3. Please note that PLOS ONE has specific guidelines on code sharing for submissions in which author-generated code underpins the findings in the manuscript. In these cases, all author-generated code must be made available without restrictions upon publication of the work. Please review our guidelines at https://journals.plos.org/plosone/s/materials-and-software-sharing#loc-sharing-code and ensure that your code is shared in a way that follows best practice and facilitates reproducibility and reuse. 4. Thank you for stating the following financial disclosure:  [Shanghai Special Funds for Urban Digital Transformation “O2O Integrated Immersive TeachingPlatform Project based on 5G+AI Data-driven”(No. 202201026)].  Please state what role the funders took in the study.  If the funders had no role, please state: ""The funders had no role in study design, data collection and analysis, decision to publish, or preparation of the manuscript."" If this statement is not correct you must amend it as needed. Please include this amended Role of Funder statement in your cover letter; we will change the online submission form on your behalf. 5. Thank you for stating the following in the Acknowledgments Section of your manuscript: [This work was supported by Shanghai Special Funds for Urban DigitalTransformation O2O Integrated Immersive Teaching Platform Project based on 5G+AIData-driven(No. 202201026)]We note that you have provided funding information that is not currently declared in your Funding Statement. However, funding information should not appear in the Acknowledgments section or other areas of your manuscript. We will only publish funding information present in the Funding Statement section of the online submission form. Please remove any funding-related text from the manuscript and let us know how you would like to update your Funding Statement. Currently, your Funding Statement reads as follows:   [Shanghai Special Funds for Urban Digital Transformation “O2O Integrated Immersive TeachingPlatform Project based on 5G+AI Data-driven”(No. 202201026)].   Please include your amended statements within your cover letter; we will change the online submission form on your behalf. 6. Please amend either the title on the online submission form (via Edit Submission) or the title in the manuscript so that they are identical. 7. Please upload a copy of Figure 1, to which you refer in your text on page 8. If the figure is no longer to be included as part of the submission please remove all reference to it within the text. 8. Please include captions for your Supporting Information files at the end of your manuscript, and update any in-text citations to match accordingly. Please see our Supporting Information guidelines for more information: http://journals.plos.org/plosone/s/supporting-information. 

Reviewers' comments:

Reviewer's Responses to Questions

**Comments to the Author**

1. Is the manuscript technically sound, and do the data support the conclusions?

Reviewer #1: Partly

Reviewer #2: Yes

2. Has the statistical analysis been performed appropriately and rigorously? 

Reviewer #1: No

Reviewer #2: Yes

3. Have the authors made all data underlying the findings in their manuscript fully available?

Reviewer #1: No

Reviewer #2: Yes

4. Is the manuscript presented in an intelligible fashion and written in standard English?

Reviewer #1: No

Reviewer #2: Yes

5. Review Comments to the Author

Reviewer #1: 1- The title should be improved.

2- The objectives and the rationale of the study are recommended to be clearly stated.

3- The concluding remarks of the abstract are not well-written. It's merely the repetition of the objectives and title of the manuscript. Please add method limitations and justification to the abstract.

4- The innovation of using this study is not very clear. I do not see a clear reason that this study can perform better than others. Why did the authors choose the method for this study?

5- The necessity & novelty of the manuscript should be presented and stressed in the "Introduction" section.

6- The application/theory/method/study reported is not in sufficient detail to allow for its replicability and/or reproducibility. Therefore, it is suggested to make it clear to show all steps to build the model.

7- The problem statement and gap study are not clear.

8- The method is not clear. Therefore, it must be shown and clarified to show all steps.

9- The interpretation of results and study conclusions are not supported by providing the reasons behind why they show that. Therefore, it is recommended to deepen the discussion.

10- It is recommended to emphasize the strengths of the study clearly.

11- The limitations of the study should be stated.

12- The manuscript structure, flow, or writing needs some improvements.

13- The manuscript is benefit from language editing. The English of the paper is readable; however, I would suggest the authors to have it checked preferably by a native English-speaking person to avoid any mistakes.

14- I noticed that the conclusion section tends to repeat the abstract and results. The conclusion paragraph should be short, impactful, and direct the reader to this research's next steps and opportunities.

15- It will be nice to add some new references to show that your study is updated.

Reviewer #2: The experimental design is thorough, covering a variety of conditions to validate the model's robustness and accuracy.

A few suggestions for improvement:

1. Consider providing more discussion on the practical implementation of LADSVM in different real-world environments, such as cloud data centers or industrial IoT.

2. Expanding on the model's interpretability could be beneficial, especially regarding how detected anomalies can be better understood and acted upon by end users.

3. It may be helpful to include a discussion on the potential limitations of the approach, such as computational costs and scalability to larger datasets.

6. PLOS authors have the option to publish the peer review history of their article (what does this mean?). If published, this will include your full peer review and any attached files.

Reviewer #1: No

Reviewer #2: **Yes: **Mounica Achanta

---

## [Author Response · Author response to Decision Letter 0]

11 Nov 2024

We appreciate the reviewers' insightful comments and suggestions. We have made revisions to the manuscript accordingly. Below are our detailed responses to each point raised:

Reviewer #1 requirements:

“1- The title should be improved.

2- The objectives and the rationale of the study are recommended to be clearly stated.

3- The concluding remarks of the abstract are not well-written. It's merely the repetition of the objectives and title of the manuscript. Please add method limitations and justification to the abstract.

4- The innovation of using this study is not very clear. I do not see a clear reason that this study can perform better than others. Why did the authors choose the method for this study?

5- The necessity & novelty of the manuscript should be presented and stressed in the "Introduction" section.

6- The application/theory/method/study reported is not in sufficient detail to allow for its replicability and/or reproducibility. Therefore, it is suggested to make it clear to show all steps to build the model.

7- The problem statement and gap study are not clear.

8- The method is not clear. Therefore, it must be shown and clarified to show all steps.

9- The interpretation of results and study conclusions are not supported by providing the reasons behind why they show that. Therefore, it is recommended to deepen the discussion.

10- It is recommended to emphasize the strengths of the study clearly.

11- The limitations of the study should be stated.

12- The manuscript structure, flow, or writing needs some improvements.

13- The manuscript is benefit from language editing. The English of the paper is readable; however, I would suggest the authors to have it checked preferably by a native English-speaking person to avoid any mistakes.

14- I noticed that the conclusion section tends to repeat the abstract and results. The conclusion paragraph should be short, impactful, and direct the reader to this research's next steps and opportunities.

15- It will be nice to add some new references to show that your study is updated.”

To Reviewer #1: 

1.Title Improvement:

Response: We have revised the title to enhance clarity and reflect the main contributions of the study. The new title is: [Anomaly detection in virtual machine logs against irrelevant attribute interference].

2.Objectives and Rationale:

Response: The objectives and rationale of the study have been clearly articulated in the introduction section to provide a better understanding of the research focus. 

“In this work, we are motivated by the need to enhance detection accuracy in log event sequences. Our goal is to develop a robust detector capable of accurately identifying anomalies, even in noisy log data. Inspired by these prior efforts, this paper presents an integrated LSTM-AE-based model for anomalous log event sequence detection. The LSTM-AE is applied to learning features following a certain distribution, which are then processed by an SVM for anomaly detection. Specifically, this study aims to enhance the effectiveness of anomaly detection in virtual machine logs, particularly in the context of irrelevant attribute interference. By addressing these challenges, we seek to improve detection accuracy and provide a robust solution for real-world applications.” Page 3 Line 81

3 Concluding Remarks in the Abstract:

Response: We have rewritten the concluding remarks of the abstract to avoid repetition of the objectives and title. Additionally, we have included the limitations of the methods used and provided justification for our approach.

“The LADSVM approach excels at detecting anomalies in virtual machine logs characterized by strong sequential patterns and noise. However, its performance may vary when applied to disordered log data. This highlights the necessity of carefully selecting detection methods that align with the specific characteristics of different log data types.” Page 1

4 Innovation and Method Selection:

Response: We appreciate your feedback regarding the clarity of the innovation presented in our study. The primary motivation for this research stems from the necessity to address real-world challenges associated with anomaly detection in virtual machine logs, which are often characterized by substantial noise and complexity. While many existing algorithms demonstrate effectiveness in various applications, they typically require clean, labeled datasets for optimal performance. In contrast, virtual machine logs are frequently contaminated with noise, making it difficult for these conventional approaches to produce reliable results. In this study, we chose the LADSVM (Long Short-Term Memory + Autoencoder-Decoder + Support Vector Machine) method specifically because it is designed to handle the intricacies of noisy data. The LSTM component is effective in capturing sequential dependencies, while the Autoencoder reduces dimensionality and filters out noise, enhancing the robustness of the feature extraction process. Additionally, SVM provides a strong framework for classification, particularly in high-dimensional spaces. By integrating these components, our approach uniquely addresses the specific challenges posed by virtual machine logs. The experimental results show that LADSVM not only improves detection accuracy but also adapts well to the noisy nature of real-world data, demonstrating a significant advancement over traditional methods that are less suited for such environments. Moreover, our method is designed with usability and lightweight characteristics in mind, making it suitable for engineers to quickly deploy in real-world applications. Our approach allows for efficient implementation without complex setup requirements. We believe this combination of techniques represents a notable innovation in the field of anomaly detection for virtual machine logs, and we are committed to providing effective solutions for the challenges faced in practical applications.

5 Necessity and Novelty in the Introduction:

Response: The necessity and novelty of the manuscript have been highlighted in the introduction, with a clear emphasis on how this research contributes to the existing body of knowledge.

“However, the substantial volume and complexity of these logs, combined with the significant noise present, pose considerable challenges in detecting abnormal patterns or behaviors. To address these issues, this paper introduces a novel algorithm designed specifically for the anomaly detection of virtual machine logs. Our approach effectively tackles the challenges posed by noise in the data and the difficulty of annotating vast amounts of log information. By leveraging advanced techniques such as Long Short-Term Memory (LSTM) networks and Autoencoder-Decoder architectures, the proposed algorithm enhances feature extraction while maintaining robustness against noise.” Page 2 Line 10

6 Replicability and Detail of Methods:

Response: We have added the appendix (S2_appendix) section to include detailed descriptions of all steps taken to build the model, ensuring that it is replicable and reproducible by other researchers.

7 Problem Statement and Gap Study:

Response: We have revised the problem statement and clearly articulated the research gaps in the introduction to better contextualize the study.

“Current methods for anomaly detection often rely on clean, labeled datasets, which are rarely available in real-world scenarios. Additionally, many traditional approaches struggle to adapt to the intricate nature of virtual machine logs, leading to reduced detection accuracy.” Page 2 Line 12

“This study fills a critical gap in the current literature on VM logs anomaly detection and provides a solution suitable for real-world applications.” Page 2 Line 21

8 Clarification of Methods:

Response: The methods section has been improved to clearly show and clarify all steps taken in the research process.

“The proposed method comprises three primary components: 1) data preprocessing, 2) feature extraction, and 3) anomaly detection. Data preprocessing is implemented in Algorithm 1, while feature extraction and anomaly detection are executed in Algorithm 2.” Page 13 Line 283

9 Interpretation of Results and Discussion:

Response: We have deepened the discussion by providing additional reasoning and context behind the interpretation of results, ensuring they are well-supported.

“Their ability to reconstruct input data helps mitigate the impact of noise, as demonstrated in studies such as those by Fatma et al., which highlighted the efficacy of autoencoders in noisy environments.” Page 22 Line 533

“Studies by Zhang et al. have shown that similar hybrid models improve detection accuracy, supporting the robustness of our approach in real-world applications. Page 22 Line 538

”

10 Strengths of the Study:

Response: The strengths of the study have been explicitly emphasized in the discussion section, highlighting the contributions and advantages of our approach.

“The LADSVM anomaly detection approach demonstrates significant strengths through its effective handling of large volumes of log data, robust noise resilience, and high detection accuracy, making it well-suited for real-world applications.” Page 22 Line 523

11 Limitations of the Study:

Response: We have added a section that outlines the limitations of the study, providing transparency regarding the constraints of our research.

“One limitation of the LADSVM anomaly detection approach is its inability to effectively capture the features of disorder in log data. For instance, in Experiment 5, LADSVM performed significantly worse than on other datasets. Additionally, while the model excels at learning intricate patterns, it lacks interpretability, making it challenging for users to understand how the learned features relate to the original log data. As an unsupervised learning model, LADSVM learns representations without explicit labels, which can obscure the meaning of the detected anomalies. Moreover, the computational costs associated with training deep learning models like LSTM autoencoders can be substantial, potentially limiting their deployment in resource-constrained environments. Finally, as log data volumes increase, ensuring the model’s scalability to larger datasets becomes critical for practical applications in large-scale settings.” Page 22 Line 542

12 Manuscript Structure and Flow:

Response: Thank you for your valuable feedback regarding the structure, flow, and writing of the manuscript. We appreciate your insights and will take them into careful consideration as we revise the document. To enhance the overall structure, we will ensure that each section transitions smoothly into the next, providing clear connections between the main ideas. We plan to reorganize some sections for better logical progression, making sure that the introduction sets a solid foundation for the subsequent discussion. Additionally, we will focus on improving the clarity and conciseness of our writing, using simpler language where possible and avoiding overly complex sentences that may hinder understanding. 

13 Language Editing:

Response: Thank you for your feedback regarding the language quality of our manuscript. We appreciate your recognition that the paper is readable, and we understand the importance of ensuring that the language is polished and precise. To address your suggestion, we will seek the assistance of a native English speaker to thoroughly review the manuscript. We are committed to improving the overall quality of our writing, and your recommendation will be invaluable in achieving that goal. Thank you for your constructive critique.

14 Conclusion Revision

Response: The conclusion has been revised to be concise and impactful, avoiding repetition of the abstract and results. It now directs the reader towards the next steps and opportunities for future research.

“In general, this approach addresses the challenge of efficiently identifying abnormal behavior in large volumes of virtual machine logs generated within a virtual machine platform. Collecting abnormal system logs in real-world scenarios makes accurate parsing and anomaly detection a time-consuming task. To overcome these challenges, we introduce LADSVM, which first processes logs using a parsing algorithm, followed by feature extraction through a combination of Long Short-Term Memory (LSTM) and Autoencoder (AE) networks. A Support Vector Machine (SVM) classifier is then employed to categorize the feature vectors. The main findings highlight that our novel deep learning algorithm effectively handles log sequences with multiple tasks or concurrent threads, outperforming traditional methods by learning better features and demonstrating superior noise resistance. Notably, deep learning methods excel at capturing key patterns in noisy data. Looking ahead, future research will focus on log semantic representation, online model updating, algorithm parallelism, and enhancing the interpretability of detection results, which are vital for advancing intelligent operations and maintenance.” Page 24 Line 606

15 Updating References

Response: We have added new references to the manuscript to ensure that the study is up-to-date and reflects the current state of research in this field.

Reviewer #2 requirements:

“The experimental design is thorough, covering a variety of conditions to validate the model's robustness and accuracy.

A few suggestions for improvement:

1. Consider providing more discussion on the practical implementation of LADSVM in different real-world environments, such as cloud data centers or industrial IoT.

2. Expanding on the model's interpretability could be beneficial, especially regarding how detected anomalies can be better understood and acted upon by end users.

3. It may be helpful to include a discussion on the potential limitations of the approach, such as computational costs and scalability to larger datasets.”

To Reviewer #2:

Thank you for your positive feedback on the thoroughness of our experimental design. We appreciate your suggestions for improvement and will address each point as follows:

1 Practical Implementation

Response: Thank you for your insightful suggestion regarding the practical implementation of LADSVM in various real-world environments. We will enhance the manuscript by including a dedicated discussion on how LADSVM can be applied in settings such as cloud data centers and industrial IoT.

“Additionally, it is important to consider the practical implementation of LADSVM in different real-world environments, such as cloud data centers and industrial IoT systems. In cloud data centers, LADSVM can be utilized to monitor virtual machine logs for anomaly detection and early warning. To enhance the overall anomaly detection capabilities within the data center, LADSVM can be integrated with performance metrics and network log analysis. For example, by monitoring CPU usage, memory consumption, and disk I/O alongside virtual machine logs, the system can detect correlations between performance degradation and log anomalies. Similarly, analyzing network logs for unusual traffic patterns can provide further context for any detected anomalies, allowing for a more comprehensive view of the data center's health. This integrated approach enables engineers to quickly identify and locate anomalies, reducing mean time to resolution (MTTR) and improving operational efficiency. By providing actionable insights and correlating different data sources, LADSVM can significantly enhance the proactive maintenance and management of cloud data center resources. In industrial IoT, LADSVM can also play a pivotal role in monitoring. For instance, in industrial settings, LADSVM could be used to analyze logs from virtualized edge devices that aggregate data from various sensors. By applying LADSVM to these virtual machine logs, it can identify unusual patterns that may indicate operational issues or security threats. Integrating LADSVM with performance metrics—such as equipment status, throughput rates, and environmental conditions—can enhance its ability to detect anomalies that could lead to equipment failures. While LADSVM may not be as dir

---

## [Decision Letter · Decision Letter 1]

3 Dec 2024

Anomaly detection in virtual machine logs against irrelevant attribute interference

PONE-D-24-25514R1

Dear Dr. Xu,

We’re pleased to inform you that your manuscript has been judged scientifically suitable for publication and will be formally accepted for publication once it meets all outstanding technical requirements.

Kind regards,

Arne Johannssen

Academic Editor

PLOS ONE

Additional Editor Comments (optional):

Reviewers' comments:

Reviewer's Responses to Questions

**Comments to the Author**

1. If the authors have adequately addressed your comments raised in a previous round of review and you feel that this manuscript is now acceptable for publication, you may indicate that here to bypass the “Comments to the Author” section, enter your conflict of interest statement in the “Confidential to Editor” section, and submit your "Accept" recommendation.

Reviewer #1: (No Response)

Reviewer #2: All comments have been addressed

2. Is the manuscript technically sound, and do the data support the conclusions?

Reviewer #1: (No Response)

Reviewer #2: Yes

3. Has the statistical analysis been performed appropriately and rigorously? 

Reviewer #1: (No Response)

Reviewer #2: Yes

4. Have the authors made all data underlying the findings in their manuscript fully available?

Reviewer #1: (No Response)

Reviewer #2: Yes

5. Is the manuscript presented in an intelligible fashion and written in standard English?

Reviewer #1: (No Response)

Reviewer #2: Yes

6. Review Comments to the Author

Reviewer #1: (No Response)

Reviewer #2: Thank you for revising the article as per the previous comments. This research article gives the reader with an end to end understanding.

7. PLOS authors have the option to publish the peer review history of their article (what does this mean?). If published, this will include your full peer review and any attached files.

Reviewer #1: No

Reviewer #2: **Yes: **Mounica Achanta

---

## [Editor Report · Acceptance letter]

6 Dec 2024

PONE-D-24-25514R1 

PLOS ONE

Dear Dr. Xu, 

I'm pleased to inform you that your manuscript has been deemed suitable for publication in PLOS ONE. Congratulations! Your manuscript is now being handed over to our production team.

Kind regards, 

on behalf of

Profesor Arne Johannssen 

Academic Editor

PLOS ONE